# Modification and analysis of context-specific genome-scale metabolic models: methane-utilizing microbial chassis as a case study

M. A. Kulyashov,[1] R. Hamilton,[2] Y. Afshin,[2] S. K. Kolmykov,[1] T. S. Sokolova,[1] T. M. Khlebodarova,[1] M. G. Kalyuzhnaya,[2] I. R. Akberdin[1]

**ABSTRACT**    Context-specific genome-scale model (CS-GSM) reconstruction is becoming an efficient strategy for integrating and cross-comparing experimental multi-scale data to explore the relationship between cellular genotypes, facilitating fundamental or applied research discoveries. However, the application of CS modeling for non-conventional microbes is still challenging. Here, we present a graphical user interface that integrates COBRApy, EscherPy, and RIPTiDe, Python-based tools within the BioUML platform, and streamlines the reconstruction and interrogation of the CS genome-scale metabolic frameworks via Jupyter Notebook. The approach was tested using -omics data collected for *Methylotuvimicrobium alcaliphilum* 20Z[R], a prominent microbial chassis for methane capturing and valorization. We optimized the previously reconstructed whole genome-scale metabolic network by adjusting the flux distribution using gene expression data. The outputs of the automatically reconstructed CS metabolic network were comparable to manually optimized *i*IA409 models for Ca-growth conditions. However, the CS model questions the reversibility of the phosphoketolase pathway and suggests higher flux via primary oxidation pathways. The model also highlighted unresolved carbon partitioning between assimilatory and catabolic pathways at the formaldehyde-formate node. Only a very few genes and only one enzyme with a predicted function in C1 metabolism, a homolog of the formaldehyde oxidation enzyme (*fae1-2*), showed a significant change in expression in La-growth conditions. The CS-GSM predictions agreed with the experimental measurements under the assumption that the Fae1-2 is a part of the tetrahydrofolate-linked pathway. The cellular roles of the tungsten (W)-dependent formate dehydrogenase (*fdhAB*) and *fae* homologs (*fae1-2* and *fae3*) were investigated via mutagenesis. The phenotype of the *fdhAB* mutant followed the model prediction. Furthermore, a more significant reduction of the biomass yield was observed during growth in La-supplemented media, confirming a higher flux through formate. *M. alcaliphilum* 20Z[R] mutants lacking *fae1-2* did not display any significant defects in methane or methanol-dependent growth. However, contrary to *fae1*, the *fae1-2* homolog failed to restore the formaldehyde-activating enzyme function in complementation tests. Overall, the presented data suggest that the developed computational workflow supports the reconstruction and validation of CS-GSM networks of non-model microbes.

**IMPORTANCE** The interrogation of various types of data is a routine strategy to explore the relationship between genotype and phenotype. An efficient approach for integrating and cross-comparing experimental multi-scale data in the context of whole-genome-based metabolic network reconstruction becomes a powerful tool that facilitates fundamental and applied research discoveries. The present study describes the reconstruction of a context-specific (CS) model for the methane-utilizing bacterium, *Methylotuvimicrobium alcaliphilum* 20Z[R]. *M. alcaliphilum* 20Z[R] is becoming an attractive microbial platform for the production of biofuels, chemicals, pharmaceuticals, and

**Peer Reviewer** Esteban Marcellin, University of Queensland, Brisbane, Queensland, Australia

Address correspondence to M. G. Kalyuzhnaya, mkalyuzhnaya@sdsu.edu, or I. R. Akberdin, akberdin.ir@talantiuspeh.ru.

M. A. Kulyashov and R. Hamilton contributed equally to this article. Author order was determined by drawing straws.

The authors declare no conflict of interest.

See the funding table on p. 14.

bio-sorbents for capturing atmospheric methane. We demonstrate that this pipeline can help reconstruct metabolic models that are similar to manually curated networks. Furthermore, the model is able to highlight previously overlooked pathways, thus advancing fundamental knowledge of non-model microbial systems or promoting their development toward biotechnological or environmental implementations.

**KEYWORDS** context-specific genome-scale metabolic modeling, methanotrophy, methane-utilizing bacteria, *Methylotuvimicrobium alcaliphilum* 20Z[R]

Complete and accurate reconstruction of metabolic networks is central for advancing fundamental discoveries or rebuilding microbial systems toward desired outcomes (1–3). Model reconstruction includes numerous steps, but in general, success relies on the efficient integration of multi-scale experimental data, such as structural and functional organization of genomes, transcriptomes, proteomes, and metabolomes. The integration requires additional mathematical model developments that can capture changes in intra- and extracellular metabolites of a cell in response to genetic and/or environmental perturbation (4). The model is expected to provide *in silico* and yet reliable solutions that reproduce the system's behaviors, such as growth rate or cell yields, and predict genetic alterations required to improve desired outcomes (i.e., product yield) under defined growth conditions (5, 6).

Advances in next-generation sequencing approaches and automatic reconstruction of genome-scale metabolic (GSM) models have led to active application of network-guided metabolic engineering of non-model organisms, often with unique metabolisms like methane utilization and lithotrophy. Numerous models are now available in BiGG (7), BioModels (8), and MetaFishNet (9). GSM reconstruction can further facilitate developments in non-model microbial systems via the addition of omics-driven constraints on the distribution of fluxes to realistically represent metabolic capabilities of the organism of interest (10, 11).

The present study describes the computational framework for the semi-automated reconstruction of a context-specific (CS) model of non-model organisms. The workflow includes a set of Python programs, including COBRApy (12), EscherPy (13), and RIPTiDe (14) integrated into the Jupyter Notebook environment on the BioUML platform (15). The pipeline was tested for the CS-GSM reconstruction of methane-utilizing bacteria (known as methanotrophs). There is a significant interest in compiling the unique metabolic capabilities of methanotrophic bacteria for biotechnological and/or environmental applications. We anticipate that the computational framework presented below can expedite the implementation of GSM models for such advancements. We selected *Methylotuvimicrobium alcaliphilum* 20Z[R], as over the years, the *Methylotuvimicrobium* spp. has become a testable microbial platform for methane capturing and valorization (16–20). The metabolic flux balance models of the whole genome have been generated and manually curated (21–23), allowing for a direct comparison between expert-curated and automated optimization of GSMs for non-model microbes.

## MATERIALS AND METHODS

### Strain, growth media, and -omics data sets

#### Strains

The lab strain of *Methylotuvimicrobium alcaliphilum* 20Z[R] is a Gram-negative, gamma proteobacterium that can grow on either methane or methanol as its main source of carbon (24). The cultures of *Methylobacterium extorquens* AM1 and *Methylobacterium extorquens* AM1 Δ*fae1* were kindly provided by Dr. N. Cecilia Martinez-Gomez (UC Berkeley) (also see Table 3). The following *Escherichia coli* strains were used for cloning and gene transfer experiments: *E. coli* S17-1 (lab stock) and *E. coli* DH10B (NEB Labs, Catalog Number C3019H): Δ(*ara-leu*)7697 *araD139 fhuA* Δ*lacX74 galK16 galE15 e14-φ80dlacZ*Δ*M15 recA1 relA1 endA1 nupG rpsL* (Str[R]) *rph spoT1* Δ(*mrr-hsdRMS-mcrBC*).

## Growth media

*M. alcaliphilum* 20Z[R] strains were routinely grown in Pi$_{3\%}$ nitrate mineral media supplemented with methanol (0.2%, vol/vol) or methane as described previously (25). Methane (25–50 mL) was introduced into 125–250 mL vials with 25–50 mL media using a syringe to keep a methane:air ratio of 20:80 in headspace. Plates were incubated in anaerobic jars and refilled using an Anoxomat system programmed to provide an atmosphere of 12%–25% methane and 75%–88% air. *M. extorquens* strains were grown using Hypho media (26), supplemented with methanol (0.2%, vol/vol) or succinate (0.4%, vol/vol). Liquid cultures were incubated at 30°C with shaking at 200 rpm. Plates were kept in 30°C incubators. *E. coli* strains were grown using LB broth or LB agar media (Miller, Difco BD Life Sciences) and incubated at 37°C. The following antibiotics were applied when required: kanamycin (100 µg/mL) and rifamycin (50 µg/mL).

## Gene expression studies

The cultivation parameters of *M. alcaliphilum* 20Z[R], culture growth rates, and sequencing details were previously described (21) and are summarized in Table 1. Briefly, the cultivation parameters and cell samples for RNA sequencing were collected from steady-state bioreactor cultures growing in nitrate mineral media at pH 8.7–9 with or without La supplementation. The parameters for each bioreactor unit (250 mL vessel with 125–150 mL of media) were set with agitation at 500 rpm, temperature kept at 30°C, and gas inflow rate of 0.3 sL/hour.

The RNA-seq reads were mapped to the *M. alcaliphilum* 20Z[R] reference genome (ASM96853v1) using the Bowtie2 algorithm version 2.5.1 (27). Illumina standard adapters were removed using Trim Galore (28) software (when necessary) before mapping. The mapped reads were quantified with FeatureCounts (29) using gene features from the RefSeq annotation (GCF_000968535.2). To extend the functional annotation of mapped genes, a comparison of three annotations (old and new RefSeq as well as Genoscope annotations) for this genome was carried out (Table S1). Subsequently, DESeq2 (30) was used to normalize and perform differential expression analysis. Genes are considered to be differentially expressed if they have |log2(fold change)| > log2(1.5) at adjusted *P*-value < 0.05. The EnhancedVolcano R-package was used to visualize the obtained results.

For KEGG signaling pathways enrichment analysis, we used a homebrew R-script. Specifically, the KEGGREST R-package was applied to match genes to KEGG signaling pathways. Then, Wilcoxon rank-sum test was performed for enrichment testing. The analysis was performed independently on either the full set of genes or specifically on genes considered in gene-protein-reaction (GPR) associations of the model.

The results of the analyses are stored on the BioUML platform (15) and available at the following link: https://gitlab.sirius-web.org/RSF/20ZR_CS_GSM_model. The RNA-seq data (counts and differentially expressed genes) were submitted to the NCBI Gene Expression Omnibus database under accession number GSE253414.

## Extension of the *i*IA409 model, flux balance analysis with COBRApy, and model visualization with EscherPy

To integrate transcriptomic data into the mathematical model, a published and experimentally verified mathematical model for 20Z[R], *i*IA409 (21) was harnessed. The model was modified to account for the presence of metals either in the medium

**TABLE 1** Summary of RNA-seq transcriptomic data sets used for the optimization of the GSM model

| Sample name | Sample ID | Raw reads | Aligned reads | Data set ID | CH$_4$/O$_2$ consumption | Growth rate (hour$^{-1}$) |
|---|---|---|---|---|---|---|
| Ca-BR1 | Ca-CH4-BR1_S1_L005_R1_001 | 22,124,770 | 10,140,552 | GSM8020389 | 1.12 ± 0.09 | 0.05 |
| CaBR2 | Ca-CH4-BR2_S1_L005_R1_001 | 17,884,237 | 16,955,446 | GSM8020390 | 1.12 ± 0.09 | 0.05 |
| LaBR1 | La-CH4-BR1_S1_L005_R1_001 | 18,542,763 | 8,409,524 | GSM8020393 | 1.28 ± 0.01 | 0.07 |
| LaBR2 | La-CH4-BR2_S1_L005_R1_001 | 16,381,413 | 12,029,600 | GSM8020394 | 1.28 ± 0.01 | 0.07 |

or transported into the cell that were not previously considered explicitly. In addition, different scenarios for the formaldehyde partitioning between tetrahydrofolate (H$_4$F) and tetrahydromethanopterin (H4MPT) pathways and phosphoketolase (EC4.1.2.22 and EC 4.1.2.9) were evaluated and compared with the original model. Modifications of the original (i.e., $i$IA409) model, as well as a subsequent flux balance analysis of the modified $i$IA409-based models and CS models reconstructed using the RIP-TiDe algorithm (14), were made via the COBRApy library (version 0.25.0 [12]) using Jupyter widgets. The program code is given at the link https://gitlab.sirius-web.org/RSF/20ZR_CS_GSM_model/-/tree/master/Data/CS_GSM_model/file_collection.files.

The EscherPy library (version 1.7.3 [13]) was employed to visualize flux distributions predicted by the modified and CS models. The final network is available at the following link: https://gitlab.sirius-web.org/RSF/20ZR_CS_GSM_model/-/tree/master/Data/CS_GSM_model/file_collection.files.

The differential reaction fluxes (DRFs) between two model predictions were visualized on the metabolic map if the ratio of fluxes for a reaction fell within the boundaries of 0.67 < DRF < 1.5 (i.e., the change in the reaction flux occurs more than 1.5 times). Additionally, reactions that were activated (On) or turned off (Off) in the CS model but not in the original were assigned to a ratio value of 1,000 (for On) or 2,000 (for Off). To visually distinguish these turned-on/off reactions from other DRFs, a specific color assignment was applied: pink for the value of 1,000 (On reactions) and aqua for the value of 2,000 (Off reactions). The visualization program code is given under the link in the Jupyter Notebook called "20Z_transcript" in sections "Escher visualization" and "DRF analysis" (https://gitlab.sirius-web.org/RSF/20ZR_CS_GSM_model/-/tree/master/Data/CS_GSM_model/file_collection.files).

## Integration of transcriptomics data into the model via RIPTiDe

The RIPTiDe algorithm (version 3.4.79 [14]) was used to integrate transcriptomic data. Counts reflecting expression levels for each gene in transcripts per million in the corresponding growth conditions (Table 1) were used as a source of transcriptomic data for the construction of CS models. Right before data integration, the selected changes were made to the model to match the selected cultivation conditions, and, if necessary, the biomass metal content was modified using the original code implemented within the Jupyter Notebook using the Jupyter widgets modules, which greatly simplified the process of the model modification. The stoichiometric coefficients for La$^{3+}$ and W$^{4+}$ in the biomass equation were set at experimentally measured concentrations (summarized in Table 2). Also, the transcriptomic data were prepared using the RIPTiDe algorithm. Because we used data normalized by DESeq2, the normalization step with RIPTiDe was skipped. The file for the algorithm was converted to TSV format, containing a column with gene IDs (the same as those used in GPR rules in the $i$IA409 model) and columns with normalized counts for each gene. For the direct reconstruction of CS models, the maxfit module was used, which analyzes models with different proportions of the original biomass in the model and selects the one in which the predicted distribution of flux had the highest level of correlation with transcriptomic data (Table 2). RIPTiDe contextualize module was applied to reanalyze steps that were not calculated by original maxfit. All codes for the model modification and transcriptomic data integration are provided within the BioUML platform and available at https://gitlab.sirius-web.org/RSF/20ZR_CS_GSM_model/-/tree/master/Data/CS_GSM_model/file_collection.files and at https://github.com/mkulyashov/20ZR_CS_GSM_model_mSystems.

## Fae phylogeny, construction of mutant strains, and phenotyping

### Phylogeny

More than 560 formaldehyde-activating enzymes (Fae) and Fae-like proteins with some homology to the Fae protein from *M. alcaliphilum* 20Z$^R$ (Gene ID: 2540616361; MEALZ_RS11875) were identified in the UniProt database. Of the 560 sequences, 78 were

**TABLE 2** Modifications and *in silico* growth conditions used for the reconstruction of CS models via RIPTiDe algorithm

| Context-specific condition | Model modification | RIPTiDe parameters |
|---|---|---|
| $CH_4$ (+Ca) | Consumption $CH_4$ −11.7 ⇒ −2.85 mmol·gDCW$^{-1}$·hour$^{-1}$ | maxfit default parameters, frac_min = 0.3 |
| | Consumption $W^{4+}$ lower bound: 0 ⇒ −1,000 | |
| | $W^{4+}$ added to biomass equation as 0.01 mmol/gDCW | |
| $CH_4$ (+La) | Consumption $CH_4$ −11.7 ⇒ −6.84 mmol·gDCW$^{-1}$·hour$^{-1}$ | maxfit default parameters, frac_min = 0.5 |
| | MXALa_for lower bound: 0 ⇒ 1.71; upper bound: 1,000 ⇒ 1.71 | |
| | FALDtpc lower bound: 0 ⇒ 5.13; upper bound: 1,000 ⇒ 5.13 | |
| | Consumption $La^{3+}$ lower bound: 0 ⇒ −1,000 | |
| | Consumption $W^{4+}$ lower bound: 0 ⇒ −1,000 | |
| | Consumption $Ca^{2+}$ lower bound: −1,000 ⇒ 0 | |
| | $Ca^{2+}$ removed from the biomass equation | |
| | $W^{4+}$ added to biomass equation as 0.01 mmol/gDCW | |
| | $La^{3+}$ added to biomass equation as 0.01 mmol/gDCW | |
| $CH_4$ (+La), Scenario 2 | Consumption $CH_4$ −11.7 ⇒ −6.84 mmol·gDCW$^{-1}$·hour$^{-1}$ | maxfit default parameters, frac_min = 0.7 |
| | MXALa_for lower bound: 0 ⇒ 1.71; upper bound: 1,000 ⇒ 1.71 | |
| | FALDtpc lower bound: 0 ⇒ 5.13; upper bound: 1,000 ⇒ 5.13 | |
| | Faeii lower bound: −1,000; upper bound: 1,000 | |
| | Consumption $La^{3+}$ lower bound: 0 ⇒ −1,000 | |
| | Consumption $W^{4+}$ lower bound: 0 ⇒ −1,000 | |
| | Consumption $Ca^{2+}$ lower bound: −1,000 ⇒ 0 | |
| | $Ca^{2+}$ removed from the biomass equation | |
| | $W^{4+}$ added to biomass equation as 0.01 mmol/gDCW | |
| | $La^{3+}$ added to biomass equation as 0.01 mmol/gDCW | |

selected for phylogenetic comparison. The sequences represent 39 species—3 archaea and 36 bacteria, including planctomycetes, firmicutes, actinobacteria, and proteobacteria, and a representative from the NC10 phylum. The evolutionary history was inferred by using the maximum likelihood method and Le and Gascuel model (31). The tree with the highest log likelihood (−12,851.33) is shown. Initial trees for the heuristic search were obtained automatically by applying Neighbor-Join and BioNJ algorithms to a matrix of pairwise distances estimated using the JTT model and then selecting the topology with superior log likelihood value. A discrete gamma distribution was used to model evolutionary rate differences among sites (five categories [+$G$, parameter = 1.2236]). Evolutionary analyses were conducted in MEGA X (32, 33).

## Complementation tests

The schematic for plasmid construction and integration of 20Z$^R$*fae* into *Methylobacterium extorquens* AM1Δ*fae1* is presented in Fig. S1. To construct *fae*-expression vectors, the *fae* genes from 20Z$^R$ and pAWP78($P_{89}$) plasmid were amplified by PCR using Flash Phusion master mix (Thermo Fisher) using *M. alcaliphilum* DNA. The *fae* genes were then integrated into the pAWP78 vector under $P_{89}$ promoter by Gibson assembly using NEBuilder (NEB Labs) and transformed into *E. coli* DH10B-competent cells. Promoter $P_{89}$ is a truncated hybrid version of the $P_{tac}$ promoter, with a fragment of the $P_{trp}$, *lac*-operator, and a mxaF ribosome binding site (AGGAAA). The promoter is known to provide constitutive expression in methylotrophic bacteria (34). The correct assemblies of the genetic constructs were validated by sequencing. The vectors were subcloned into *E. coli* S17-1. The plasmids were then transferred via conjugation into an *M. extorquens* AM1Δ*fae1* mutant via bi-parental mating. pAWP78 empty vector was cloned into wild-type *M. extorquens* AM1 and the mutant strain AM1Δ*fae1* to construct control variants (Table 3). Lawns of *M. extorquens* AM1Δ*fae1* supplemented with succinate (1–2 day growth) and lawns of pAWP78- $P_{89}$-20Z$^R$*fae1* in *E. coli* S17-1 (overnight) were prepared for mating on plates with Hypho media supplemented with 5% Nutrient Broth (NB Difco, BD Life Sciences). Plates were incubated for 1 day at 30°C. The biomass

**TABLE 3** Strains used in this study

| Strain | Description | Reference/source |
|---|---|---|
| AM1 WT | *Methylobacterium extorquens* AM1 wild type | Dr. N. C. Martinez-Gomes (UC Berkeley) |
| AM1Δ*fae1* | *M. extorquens* AM1 lacking *fae1* gene | Dr. N. C. Martinez-Gomes (UC Berkeley) (35) |
| AM1-WT- EV | *M. extorquens* AM1 wild type harboring pAWP78 empty vector. Kan$^R$ | This study |
| AM1Δ*fae1*-EV | *M. extorquens* AM1Δ*fae1* harboring pAWP78 empty vector. Kan$^R$ | This study |
| AM1Δ*fae:*20Z$^R$*fae* | *M. extorquens* AM1Δ*fae* harboring pAWP79-P$_{89}$-20Z$^R$*fae1*. Kan$^R$ | This study |
| AM1Δ*fae:*20Z$^R$*fae1-2* | *M. extorquens* AM1Δ*fae* harboring pAWP78-P$_{89}$-20Z$^R$*fae1-2*. Kan$^R$ | This study |
| AM1Δ*fae:*20Z$^R$*fae3* | *M. extorquens* AM1Δ*fae* harboring pAWP78-P$_{89}$-20Z$^R$*fae3*. Kan$^R$ | This study |
| 20Z$^R$ | *Methylotuvimicrobium alcaliphilum* 20Z$^R$ WT, selected rifamycin-resistant variant | Kalyuzhnaya Lab (SDSU) |
| 20Z$^R$Δ*fae1* | *M. alcaliphilum* 20Z$^R$ lacking canonical *fae* gene | This study |
| 20Z$^R$Δ*fae1-2* | *M. alcaliphilum* 20Z$^R$ lacking *fae1-2* gene, a homolog of *fae* | This study |
| 20Z$^R$Δ*fae3* | *M. alcaliphilum* 20Z$^R$ lacking *fae3* gene | This study |
| 20Z$^R$Δ*fdhAB* | *M. alcaliphilum* 20Z$^R$ lacking canonical *fae* gene | This study |
| DH10B | Chemically competent *E. coli* cells, used for gene cloning | NEB Labs (https://www.neb.com) |
| S17-1 | Chemically competent *E. coli* cells, used for conjugation | NEB Labs (https://www.neb.com) |

was collected and spread onto Hypho-agar plates supplemented with succinate and kanamycin for selection. Kanamycin-resistant clones were observed after 3–4 days of incubation. Clones were transferred into Hypho-agar plates with kanamycin and rifamycin for *E. coli* counter selection.

The *M. extorquens* AM1Δ*fae1*-mutant complementation tests were conducted using (i) control: Hypho medium supplemented with succinate (0.4%); (ii) sensitivity test: Hypho medium supplemented with succinate (0.4%) and methanol (0.2%); (iii) growth test: Hypho medium supplemented with methanol (0.2%). Cultures were incubated at 30°C and evaluated after 1–2 days (succinate plates) and after 3–4 days (methanol or methanol + succinate plates).

Liquid batch culture experiments were carried out in flasks containing 25 mL of Hypho and supplemented with succinate (0.4%) and/or methanol (0.2%). At least three biological replicates with two technical replicates each per culture were tested. Each biological replicate was inoculated from a single colony. Growth measurements were taken starting from OD 0.1 until reaching the stationary phase and were used to calculate growth rates, and standard deviations were calculated to produce a graphic representation of the acquired data.

### Mutagenesis

All mutants were constructed using the allelic exchange vector pCM433 (36). Upstream and downstream flanking regions of *fae* genes or *fdhAB*-gene cluster (MEALZ_1882-MEALZ_1883) genes from *M. alcaliphilum* 20Z$^R$ were amplified using 2× Fusion Phlash (Thermo Fisher). The pCM433 vector was amplified in the same way using pCM433-F and pCM433-R primers (25). The PCR products were run on a gel and purified. The flanking regions were integrated into pCM433 vector using a NEBuilder HiFi DNA assembly master mix. Reactions were transformed into *E. coli* DH10B-competent cells. Kanamycin-resistant colonies were selected. Plasmids were purified using Zymoclean Gel DNA recovery kit (Zymo Research, USA). The constructs were verified by PCR, and sequences were confirmed by sequencing. The resulted plasmids were transformed into *E. coli* S17-1 and mated with *M. alcaliphilum* 20Z$^R$. Kanamycin-resistant colonies were subjected for negative selection on sucrose. The genotype of resulted clones was confirmed by PCR and PCR-fragment sequencing. All constructed strains are listed in Table 3.

## RESULTS

Below, we summarize the main steps of the reconstruction and analysis of a CS GSM (Fig. 1) using the previously developed GSM model of *M. alcaliphilum* 20Z$^R$ (21) and the RIPTiDe algorithm (14).

## Construction of CS-GSM model for aerobic methanotroph *M. alcaliphilum* 20Z^R

To build a workflow for the CS modeling with GSM and available -omics data, we applied Jupyter Notebook (37, 38). The computational platform included Python-based tools such as COBRApy (12), EscherPy (13), and RIPTiDe (14) to simplify the process of handling GSM models.

To reconstruct CS-GSMs for *M. alcaliphilum* 20Z^R, the original GSM model was extended. Specifically, the Ca^{2+} transport reactions were added as an environmental control of the enzymatic reaction catalyzed by Ca^{2+}-dependent methanol dehydrogenase (Ca-MDH, *mxaFI*) (Fig. 2A). Similarly, the transport reaction for La^{3+} was added and linked with the La-dependent methanol dehydrogenase (La-MDH, *xoxF*) reaction (Fig. 2B). Since it has been shown that La-MDHs can oxidize methanol to formate (39), the corresponding reaction was added in addition to the methanol to formaldehyde conversion. These modifications enabled tunable control of the methanol conversion steps depending on the cultivation conditions. This modification also allows control of the methanol flux via Ca-MDH or La-MDH according to their gene expression levels. Additionally, a transport reaction for tungsten (W) was added and integrated with the reaction catalyzed by W-dependent formate dehydrogenase (Fig. 2C).

The growth rates predicted by the *i*IA409 and optimized CS models were similar, but both were below the experimentally observed rates for Ca-growth conditions (Table S2). However, the predicted $O_2:CH_4$ ratios were similar to the experimental measurements (1.22 and 1.273 for *i*IA409 and CS model, respectively, versus 1.12 ± 0.09 obtained for chemostat cultures).

Similarly to the original model, the predicted $O_2$ consumption reaction rate was significantly higher than experimental measurements for the La-growth conditions too (see details in Table S2). The introduction of $O_2$ uptake constraints to the CS model tuned the *in silico* prediction toward measured parameters. Despite similar outputs, some differences between the two types of *M. alcaliphilum* genome-scale metabolic models were observed, in particular, differences in predicted growth rates, suggesting variations in carbon flux distributions. To investigate this further, we carried out a comparative analysis of the fluxes and applied network visualization (Escher tool) to highlight the most significant divergences.

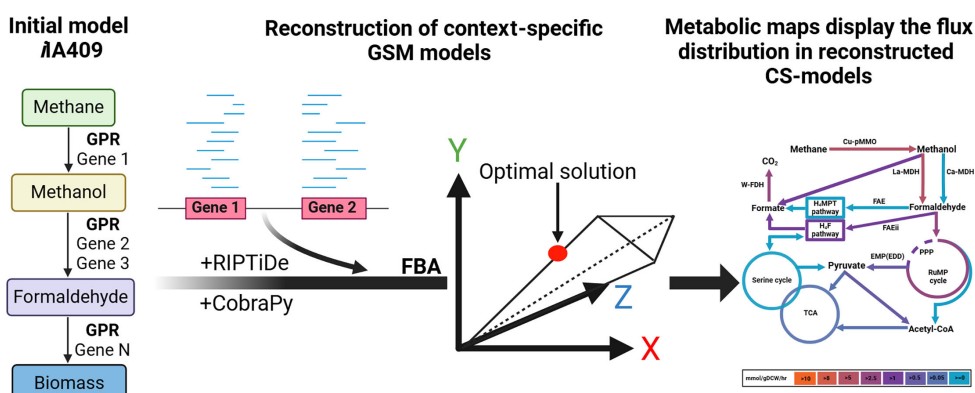

**FIG 1** A schematic representation of the developed pipeline to reconstruct and analyze the CS GSM model for *M. alcaliphilum* 20Z^R. At the first step, the original model was extended using COBRApy (12) to account for the presence in the medium and transport into the cell of some metals (see Materials and Methods). Thereafter, the transcriptomics data for different growth conditions (or contexts) were integrated into the extended model to reconstruct a number of CS models using the RIPTiDe tool (14) based on the GPR presented in the model. Finally, flux balance analysis (FBA) was conducted in COBRApy, and predicted distributions of fluxes were displayed on the metabolic maps using the Escher web tool (13). All steps of the pipeline are performed in the corresponding Jupyter Notebook (see Data availability statement).

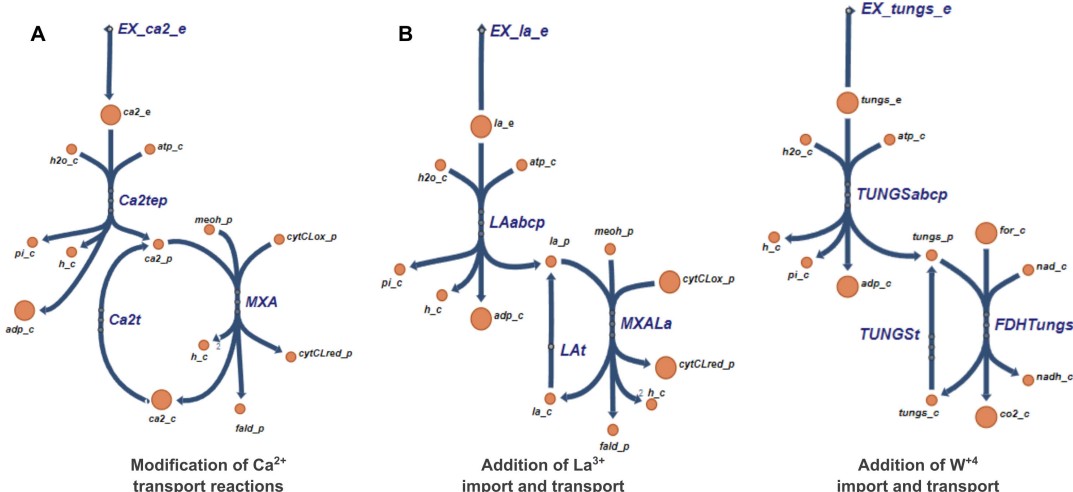

**FIG 2** Visualization of the reactions that were added to the model for calcium (A), lanthanum (B), and tungsten (C) via Escher tool (13). (A) Modified $Ca^{2+}$ transport reactions. (B) Added exchange and transport reactions for $La^{3+}$. (C) Added exchange and transport reactions for $W^{4+}$.

## CS-GMS highlights unresolved carbon distribution during growth in the presence of lanthanides

The Ca- and La-growth conditions were investigated separately. Detailed Escher metabolic maps are presented in Fig. S1.2 and 1.3 for Ca-growth condition and in Fig. S1.5 and 1.6 for La growth condition. The CS model predicts a slightly higher activity of the H4MPT pathway (from 1.07 to 1.25 mmol/gDCW/hour compared to the original model) and activation of the uphill electron transport mechanism unlike the original model for Ca-growth (Fig. S1.2 to 1.4).

The flux distribution predicted by the CS model for growth with La differed more substantially from that predicted by the original model. To quantitatively identify metabolic differences between the original model and the CS model, the metabolic map was constructed using DRFs (see Materials and Methods). The constructed map for La growth conditions showed differences in the value of fluxes of reactions related to the RuMP pathway: reactions associated with ribulose-5-phosphate regeneration are less active in the CS model simulation (decreased flux through transaldolase and transketolase reactions [TKT1 and TKT2] by 1.37 times, from 1.28 to 0.933 and 1.31 to 0.952 mmol/gDCW/hour, respectively). A similar trend was observed for pentose-5-phosphate 3-epimerase reaction from 2.59 to 1.88 mmol/gDCW/hour. Carbon flux via the EMP and EDD pathways also decreased compared to the original model predictions by 1.37 times. The decreased flux of $CO_2$ production reaction (by 1.67 times, from 3.55 to 2 mmol/gDCW/hour) was accompanied by lower $CO_2$ production via the formate dehydrogenase (FDH) reaction by 1.9 times (from 2.95 to 1.56 mmol/gDCW/hour), similar to the Ca-growth condition but to a higher degree (Fig. S1.5 to 1.7).

At the same time, the carbon flow via the C1 transfer pathways was increased. The high flux via the $H_4$MTP pathway can be linked with a very high-level expression of *fae1-2,* a homolog of canonical FAE, also known as 5,6,7,8-tetrahydromethanopterin hydro-lyase (35). *fae1-2* expression was more than four times higher during La-supplemented growth, while the expression of other genes in the $H_4$MPT pathway did not significantly change (Fig. S1.8; Table 4). Furthermore, a twofold reduction was observed for the canonical FAE1. The differential expression of the *fae* genes between Ca and La growth conditions has been previously noted, and while the alternative metabolic function for *fae1-2* has been speculated, it remains unresolved (21).

To evaluate the *fae* homolog's contribution to carbon flux deterring from the RuMP pathway toward the primary oxidation pathway, we reconstructed the CS model, which assumes different roles for the Fae1-2, from its contribution to Fae1 function to the

carbon flow into the $H_4F$ pathway by adding the reaction FAEii, which is the condensation reaction of formaldehyde with tetrahydrofolate and considering different ratios of formate and formaldehyde produced by La-MDH (*xoxF*) (Fig. 3; Table 5).

We also investigated an opportunity of the reverse function for Fae1-2 (hydrolysis instead of condensation). The assumption of Fae1-2 role in the $H_4Folate$ pathway provided the best fit to the experimental data (Table 5) for both the original and CS models. Furthermore, the CS model in this condition or the context predicts $O_2$ consumption rate without any restrictions on its reactions, unlike CS model without FAEii reaction. Considering that La-growth contributes to a higher formate pool, the essentiality of the Fae1-2 function becomes more apparent. To further investigate the prediction, additional targeted mutagenesis studies were carried out. First, we investigated the impact of the formate dehydrogenase knockout on cell growth with Ca versus La. Then, we investigated the phenotypes of *fae1-2* knockouts. The functional characterization of Fae1-2 was supplemented by phylogenetic and complementation studies.

## Mutagenesis

A knockout strain 20Z$^R$ Δ*fdhAB* was constructed as described in Materials and Methods. The growth phenotypes of the mutant were tested using growth media containing either La or Ca to investigate the impacts of the La-switch (i.e., XoxF methanol dehydrogenase instead of Ca-dependent MxaFI). The 20Z$^R$Δ*fdh* knockouts displayed growth defects when compared to the wild type, and the growth reduction was more severe during La supplementation (Fig. 4). The growth rate was almost fivefold lower in the *fdhAB* mutant compared to the wild type. No significant growth defects were observed in media supplemented with methanol (data not shown), suggesting that the FDH is more critical for methane utilization.

We also constructed *M. alcaliphilum* 20Z$^R$ mutants lacking *fae1-2* or *fae3*; however, the mutants did not display any growth defects (data not shown). We were not able to generate *fae1* knockouts, which indicates its essentiality for the growth.

## Complementation tests

In order to investigate the functional activity of the Fae1 and Fae1-2 homologs, complementation studies were carried out. *M. extorquens* AM1Δ*fae1,* the null mutant strain, does not have the ability to use methanol as a carbon source and is highly sensitive to methanol (35). The strain AM1 has been used for investigating the function of Fae-like sequences obtained from pure cultures as well as metagenomic sequences (40, 41). In this study, three plasmids, harboring *fae1* (MEALZ_RS11875)*, fae1-2* (MEALZ_RS04100), and *fae3* (MEALZ_RS07105) homologs from *M. alcaliphilum* 20Z$^R$, were constructed and integrated into *M. extorquens* AM1Δ*fae1*. All strains have the ability to grow on succinate (Fig. 5A and B). When tested on solid media, both the sensitivity test and the growth test revealed that the integration of *fae1* and *fae1-2* homologs into AM1Δ*fae1* relieved methanol sensitivity through complementation. The strain AM1Δ*fae1*:pAWP78-P$_{89}$-20Z$^R$*fae*3 displayed similar sensitivity to methanol as the parental AM1Δ*fae1* strain, while both AM1Δ*fae1*: pAWP78-P$_{89}$-20Z$^R$*fae1* and AM1Δ*fae1*:pAWP78-P$_{89}$-20Z$^R$*fae1*-2 became resistant to methanol. The *fae1* and *fae1-2* genes also rescued the AM1Δ*fae1* growth on methanol on solid media.

The growth of the strains was also investigated in liquid culture and produced surprisingly different outcome for the Fae1-2 construct. While both *fae1* and *fae1-2*

**TABLE 4** Comparison of transcription level for FAE genes (*fae1, fae1-2,* and *fae3*) between Ca- and La-present conditions

| Gene ID | Number of normalized counts, +Ca | | Number of normalized counts, +La | | log2fold change | $P_{adj}$ |
|---|---|---|---|---|---|---|
| MEALZ_RS11875 (*fae1*) | 22,233 | 21,377 | 14,086 | 13,351 | −0.431 | 0.001 |
| MEALZ_RS04100 (*fae1-2*) | 2,560 | 1,834 | 40,690 | 24,419 | 4.094 | 7.01E$^{-17}$ |
| MEALZ_RS07105 (*fae3*) | 6,392 | 5,327 | 3,519 | 3,245 | −0.55 | 0.00026 |

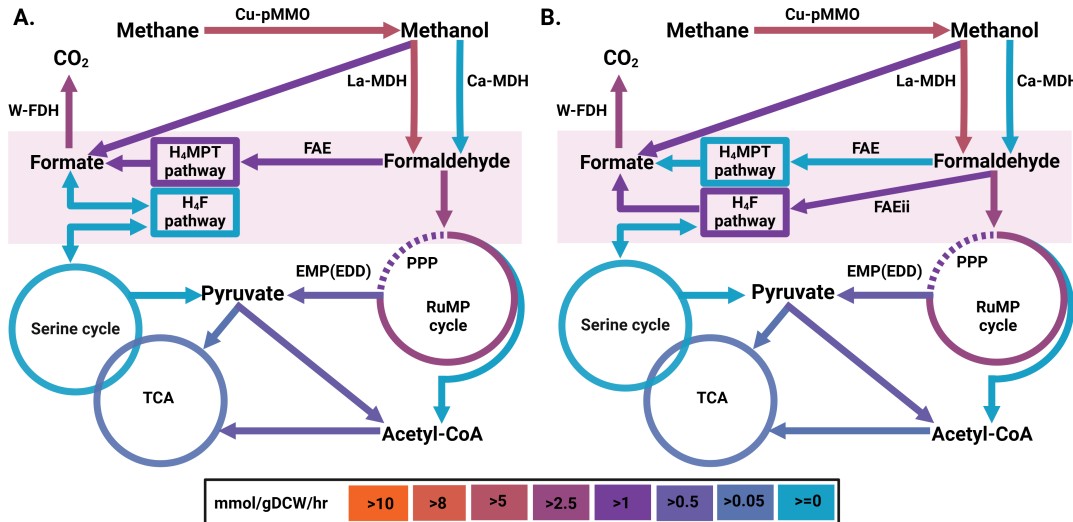

**FIG 3** The schematic representation of the distribution of fluxes predicted (A) by original *i*IA409 model for growth on $CH_4$ in the presence of La, W, and Cu with FAEii in H4MTP mode and (B) by CS model based on transcriptomic count data for growth on $CH_4$ in the presence of La, W, and Cu with FAEii in H4Folate mode. The color indicates the extent of the flux rate through the reaction in mmol/gDCW/hour. Cu-pMMO, a reaction catalyzed by Cu-dependent pMMO; La-MDH and Ca-MDH, reactions with activities of two MDH isoforms (La- and Ca-dependent, respectively); W-FDH, the enzymatic reaction catalyzed by W-dependent FDH. FAEii, reaction that describes the potential role of *fae1-2* in the condensation of formaldehyde and tetrahydrofolate.

restored growth on methanol on solid plates, only *fae1* is able to restore the AM1Δ*fae*-mutant phenotype. AM1Δ*fae*:20Z*fae1-2* is able to relieve methanol sensitivity on agar medium but is not able to grow in liquid culture supplemented with methanol (Fig. 5C through E). AM1Δ*fae1*, AM1Δ*fae*:20Z^R*fae1-2*, and AM1Δ*fae1*:20Z^R*fae3* do not grow in liquid cultures supplemented with methanol.

## Phylogenetic analyses of Fae homologs

To analyze the functional roles of the *fae* homologs, and more specifically the La-induced *fae1-2* gene product, we carried out phylogenetic analyses. Fae and Fae-like proteins are

**TABLE 5** Predicted flux via the cofactor ($H_4F$ or $H_4MTP$) mediated $C_1$ transfer for growth with La

| Scenario | MXA-La type | Flux through FAEii | Model growth rate (hour$^{-1}$) | Model $O_2$:$CH_4$ ratio | Experimental growth rate (hour$^{-1}$) | Experimental $O_2$:$CH_4$ ratio |
|---|---|---|---|---|---|---|
| *i*IA409 H4MTP | MXALa:MXALa_for = 0.75:0.25[a] | 1.29 | 0.74 | 1.3 | 0.07 | 1.28 ± 0.01 |
|  | MXALa:MXALa_for = 1.0:0.0[b] | 2.57 | 0.0822 | 1.22 | | |
| *i*IA409 H4F | MXALa:MXALa_for 0.75:0.25[a] | 1.17 | 0.0763 | 1.285 | | |
|  | MXALa:MXALa_for = 1.0:0.0[b] | 2.33 | 0.0869 | 1.185 | | |
| CS H4MTP | MXALa:MXALa_for = 0.75:0.25[a] | 2.34 | 0.0537 | 1.31 | | |
|  | MXALa:MXALa_for = 1.0:0.0[b] | 2.78 | 0.0778 | 1.27 | | |
| CS H4F | MXALa:MXALa_for = 0.75:0.25[a] | 1.17 | 0.0763 | 1.285 | | |
|  | MXALa:MXALa_for = 1.0:0.0[b] | 2.59 | 0.0657 | 1.17 | | |

[a]MXALa_for bounded conditions, in which the proportion of formate and formaldehyde produced by La-MDH (*xoxF*) was added manually by constriction of flux through MXALa and MXALa_for reactions in proportion 0.75:0.25.
[b]MXALa_for unbounded conditions, in which proportion of formate and formaldehyde produced by La-MDH (*xoxF*) was not manually restricted and bounds for MXALa and MXALa_for reactions were the same (0;1,000 mmol/gDCW) for both.

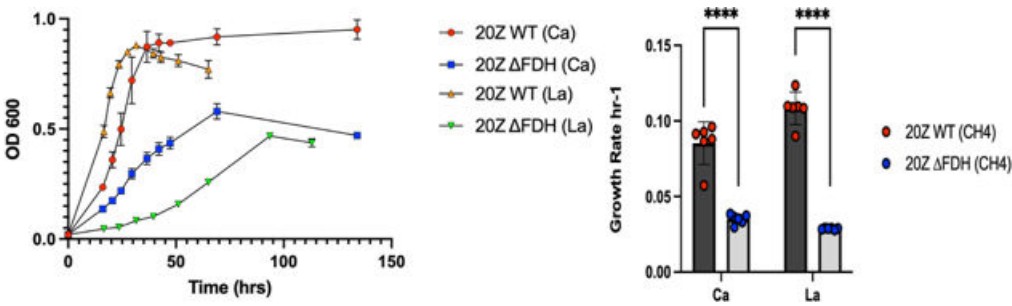

FIG 4 Growth parameters of *M. alcaliphilum* 20Z[R] WT and 20Z[R]Δ*fdhAB* grown with methane using growth media supplemented with Ca or La.

homologous across different groups of bacteria, such as the firmicutes, actinomycetes, planctomycetes proteobacteria, and NC10 phylum, as well as archaea. The phylogenetic tree we constructed includes 78 *Fae* sequences that represent different microbial species, including 34 bacterial and 5 archaeal species. *Geoglobus ahangarhi* (Gene ID: 2810347888; Ga0325130_111788) roots the tree. Five main groups were identified. Only sequences from group 1 (bacterial Fae) and group 5 (archaeal Fae) have defined cellular function (Fig. 6). Group 1 Fae proteins capture formaldehyde generated from primary $C_1$ oxidation (methanol to formaldehyde). Group 5 Fae enzymes contribute to capturing formaldehyde produced by a reverse hexulose phosphate synthase activity during the conversion of C6 sugars into C5 sugars for nucleotide biosynthesis (42, 43). Both Group 1 and Group 5 Fae proteins catalyze the formation of methylene-$H_4$MPT from $H_4$MPT and

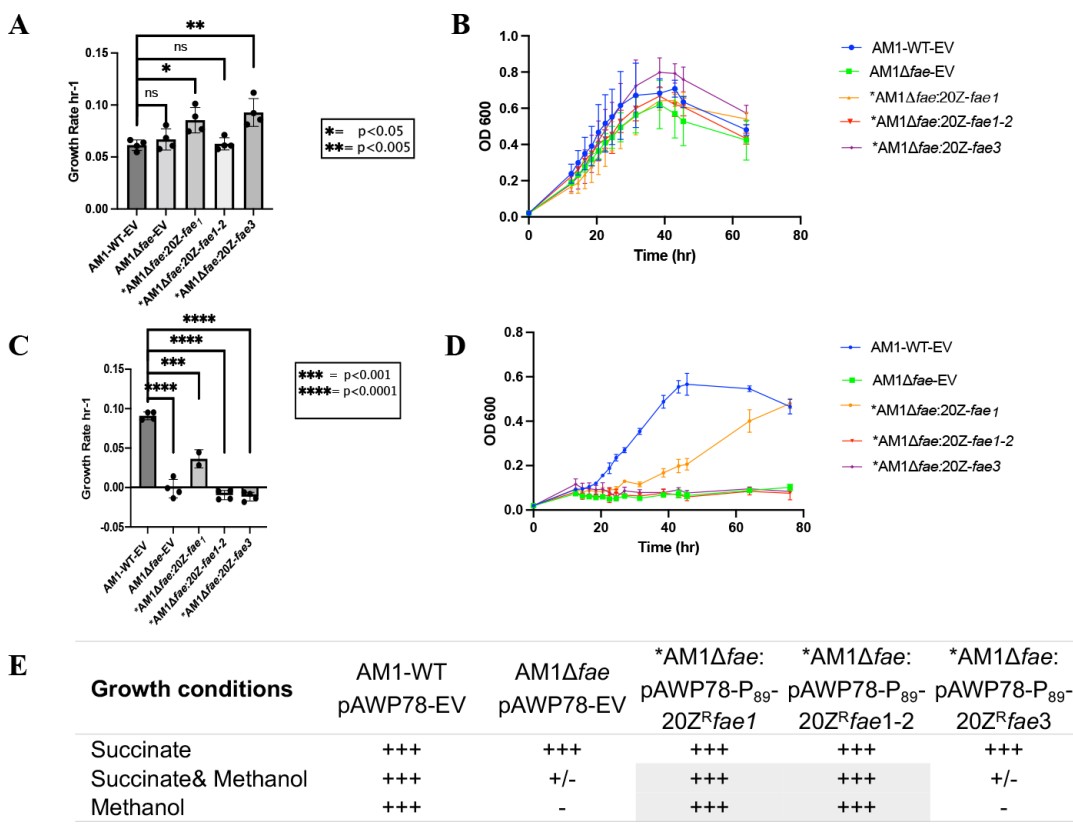

FIG 5 Growth parameters (growth rate and growth curve) of the *M. extorquens* AM1 wild-type (WT) and AM1Δ*fae* strains expressing *fae1* and *fae* homologs on succinate (A and B) and methanol (C and D). (E) Complementation results obtained on solid media. *Complementation of the *AM1Δfae*-mutant growth on solid media with methanol is highlighted in gray.

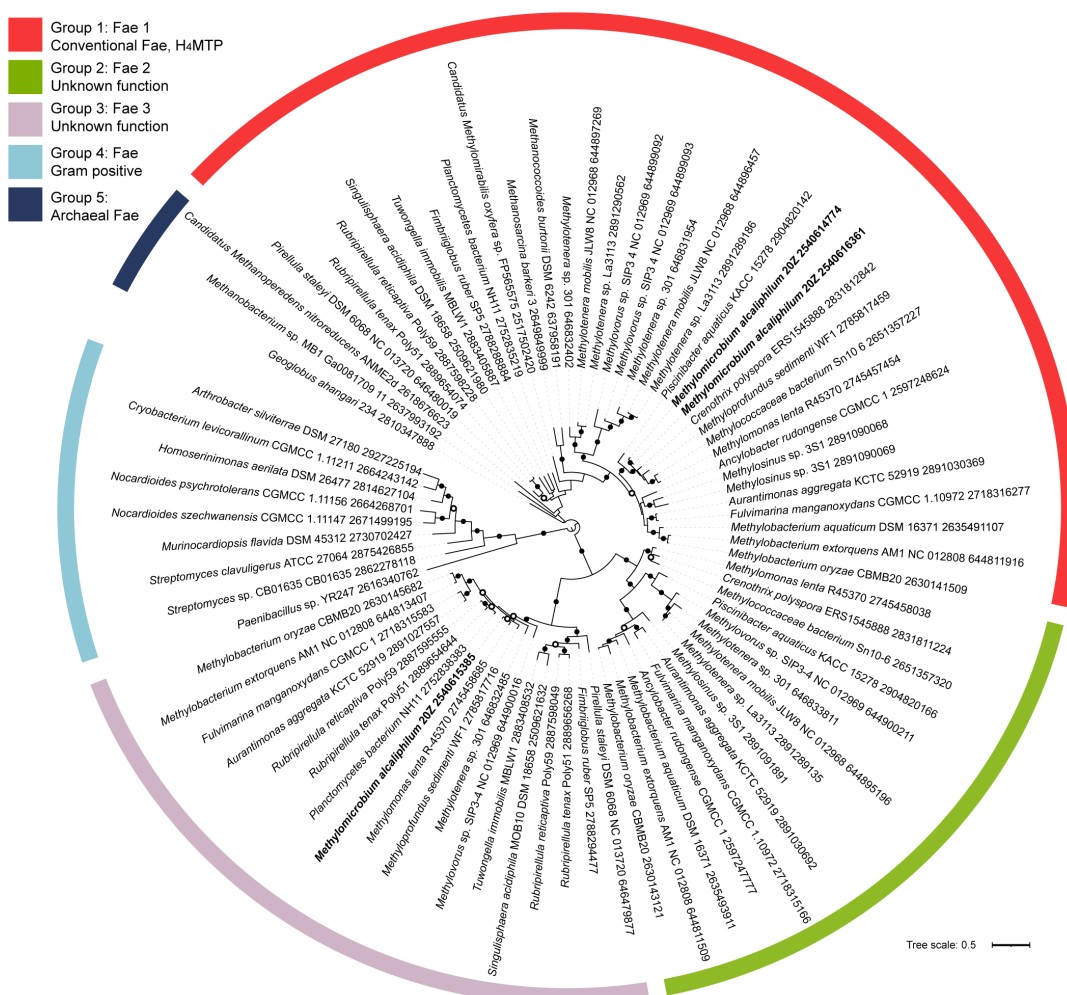

**FIG 6** Phylogenetic analysis of Fae homologs in bacteria and archaea. Protein sequences of Fae from *M. alcaliphilum* 20Z[R] are highlighted in bold. Filled circles on nodes represent >70% bootstrap support. Open circles on nodes represent >50% bootstrap support. The tree is drawn to scale, with branch lengths measured in the number of substitutions per site. This analysis involved 78 amino acid sequences. A total of 452 positions were used in the final data set.

formaldehyde and thus serve as the entry point for the formaldehyde oxidation pathway. The percent identity shared for archaeal sequences in Group 5 with canonical Fae from *M. extorquens* AM1 (Gene ID: 644811916; MexAM1_META1p1766) is between 50% and 51%.

The cellular functions of Fae-like proteins from Groups 2, 3, and 4 are not known. Suggested functions include regulatory or sensing roles or a possible contribution to tetrahydrofolate-linked C-transfer (44, 45).

## DISCUSSION AND CONCLUSIONS

In this study, we implemented the Python-based pipeline to reconstruct the context-specific metabolic network of *M. alcaliphilum* 20Z[R], a model methanotrophic bacterium. The initial goal of the study was to streamline metabolic network reconstruction for non-model microbial systems with unique metabolic capabilities, i.e., methanotrophy. The model reconstruction was followed by flux balance analysis for 20Z[R] growing on methane with Ca or La supplementation. Considering that a great deal of information is available for the strain, the reconstructed model was compared to previously expert-curated reconstructions (*i*IA409 model [21]).

The CS model predictions were comparable to the original *i*IA409 model simulation with a few exceptions. The CS model indicated that reverse electron transfer through the cytochrome bc1 complex is essential for supporting methane oxidation, whereas the original model suggested a more direct redox input from a reduced cytochrome cL, a product of methanol oxidation. Therefore, the CS model predicted the necessity of the reverse electron transfer systems without requiring additional expert-curated constraints, which were necessary for the original model (46). It should be mentioned, however, that both models do not support the redox model of methane oxidation. The CS model also ruled out a possibility of carbon flux from acetyl-P toward pentose phosphate pathway intermediates via the phosphoketolase pathway. Finally, CS model suggested higher carbon flux via formaldehyde-formate nodes. The direct conversion to formate due to XoxF activity and the $H_4$MPT pathway activation led to the enhanced production of formate compared to the value predicted by the original model. The increase in the expression of the gene encoding FDH has been confirmed by the analysis of DEGs between La and Ca conditions (47). Follow-up studies with the formate dehydrogenase knockouts further confirm the essentiality of the formate oxidation reaction.

The CS model predictions for the La-dependent growth condition demonstrated a set of metabolic rearrangements in comparison to the flux distributions obtained by the simulation of the original model. The key difference in the carbon flux distribution predicted by the CS versus the original models is the increased role of the C1 transfer pathway. It has been shown previously that La supplementation results in the reduction of gene expression for a canonical *fae1,* while inducing its close homolog, named here as *fae1-2*. We interrogated the possible functional role of *fae1-2* using the CS model. The model highlights a possibility that Fae1-2 is an enzyme that contributes to formaldehyde condensation with $H_4$folate. Genetic tests further indicate that the Fae1-2 function differs substantially from the canonical Fae1.

In summary, we demonstrate that this pipeline can help reconstruct metabolic models that are similar to manually curated networks. The pipeline for CS model development implemented in the BioUML platform can be used and easily tuned for the investigation of metabolic changes in other growth conditions of the methanotrophic strain. Furthermore, the reconstructed models should be able to highlight previously overlooked pathways, thus advancing fundamental knowledge of non-model microbial systems and promoting their development toward biotechnological or environmental implementations.

## ACKNOWLEDGMENTS

The authors thank Dr. Nathalie Delherbe (SDSU, Biology) for her recommendations and support of the phylogenetic analysis presented in this study.

The study was financially supported by the Russian Science Foundation (project No. 23-24-00606, https://rscf.ru/en/project/23-24-00606/) and by the Department of Energy (USA) Award Number: DE-SC0019181.

I.R.A. and M.G.K. designed and coordinated the study. M.A.K. reconstructed the CS GSM model for *M. alcaliphilum* 20Z[R] and conducted FBA analysis of the updated model. R.H. carried out fermentation, RNA-seq studies, and *fdhAB* mutagenesis. Y.A. carried out *fae* mutagenesis and complementation studies. T.S.S. and S.K.K. conducted RNA-seq analysis. M.A.K., T.M.K., I.R.A., and M.G.K. analyzed the data and FBA results. M.A.K., S.K.K., T.M.K., I.R.A., and M.G.K. wrote the manuscript. All authors read and approved the final manuscript.

## AUTHOR AFFILIATIONS

[1]Department of Computational Biology, Scientific Center for Genetics and Life Sciences, Sirius University of Science and Technology, Sochi, Russia

[2]Department of Biology and Viral Information Institute, San Diego State University, San Diego, California, USA

## AUTHOR ORCIDs

M. G. Kalyuzhnaya http://orcid.org/0000-0002-9058-7794
I. R. Akberdin http://orcid.org/0000-0003-0010-8620

## FUNDING

| Funder | Grant(s) | Author(s) |
|---|---|---|
| Russian Science Foundation (RSF) | 23-24-00606 | M. A. Kulyashov |
| | | S. K. Kolmykov |
| | | T. S. Sokolova |
| | | T. M. Khlebodarova |
| | | I. R. Akberdin |
| U.S. Department of Energy (DOE) | DE-SC0019181 | R. Hamilton |
| | | Y. Afshin |
| | | M. G. Kalyuzhnaya |

## AUTHOR CONTRIBUTIONS

M. A. Kulyashov, Formal analysis, Investigation, Methodology, Software, Visualization, Writing – original draft | R. Hamilton, Formal analysis, Investigation, Methodology, Visualization | Y. Afshin, Investigation, Methodology, Visualization | S. K. Kolmykov, Formal analysis, Investigation, Methodology, Software, Writing – original draft | T. S. Sokolova, Formal analysis, Investigation, Methodology, Visualization | T. M. Khlebodarova, Data curation, Formal analysis, Investigation, Writing – original draft | M. G. Kalyuzhnaya, Conceptualization, Funding acquisition, Investigation, Project administration, Supervision, Writing – original draft, Writing – review and editing | I. R. Akberdin, Conceptualization, Funding acquisition, Investigation, Project administration, Supervision, Writing – original draft, Writing – review and editing

## DATA AVAILABILITY

The RNA-seq data (counts and differentially expressed genes) are available via the GEO database under accession number GSE253414. The described pipeline for transcriptomics data analysis, results of the DEG analysis, and the CS GSM model for 20ZR with the corresponding Jupyter Notebook are available on Gitlab at https://gitlab.sirius-web.org/RSF/20ZR_CS_GSM_model (accessed on 14 August 2024) and on Github at https://github.com/mkulyashov/20ZR_CS_GSM_model_mSystems (accessed on 3 December 2024).

## ADDITIONAL FILES

The following material is available online.

### Supplemental Material

**Supplemental Figures (mSystems01105-24-s0001.docx).** Fig. 1.1-1.8.
**Table S1 (mSystems01105-24-s0002.xlsx).** The functional annotation of mapped genes and a comparison of two annotations: Refseq (old and new RefSeq IDs) with Genoscope annotations for 20Z$^R$ genome.
**Table S2 (mSystems01105-24-s0003.xlsx).** Simulation results of the original iIA409 model and CS-GSMs for different growth conditions.
**Table S3 (mSystems01105-24-s0004.xlsx).** Results of the comparative analysis for model predictions on fluxes distribution versus transcriptomics and proteomics data.

Open Peer Review

**PEER REVIEW HISTORY (review-history.pdf).** An accounting of the reviewer comments and feedback.

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
