## [Reviewer comments · mSystems]

Modification and Analysis of Context-Specific Genome-Scale Metabolic Models: Methane-Utilizing Microbial Chassis as a Case Study

Mikhail Kulyashov, Richard Hamilton, Yasmine Afshin, Semyon Kolmykov, Tatyana Sokolova, Tamara Khlebodarova, Marina Kalyuzhnaya, and Ilya Akberdin

Corresponding Author(s): Marina Kalyuzhnaya, San Diego State University

Review Timeline:

Submission Date:	September 11, 2024
Editorial Decision:	November 16, 2024
Revision Received:	November 26, 2024
Accepted:	December 2, 2024

Editor: Oleg Igoshin

Reviewer(s): Disclosure of reviewer identity is with reference to reviewer comments included in decision letter(s). The following individuals involved in review of your submission have agreed to reveal their identity: Esteban Marcellin (Reviewer #3)

Transaction Report:

DOI: <https://doi.org/10.1128/msystems.01105-24>

Re: mSystems01105-24 (Modification and Analysis of Context-Specific Genome-Scale Metabolic Models: Methane-Utilizing Microbial Chassis as a Case Study)

Dear Dr. Marina G. Kalyuzhnaya:

Revision Guidelines

Sincerely,
Oleg Igoshin
Editor
mSystems

Reviewer #1 (Comments for the Author):

There are still inconsistencies in the naming conventions of fae - namely 1, 2, 3 or 1-2, 3.

In addition, there are inconsistencies between the legends and what is presented in figures. The line thickness only comes across in the supplemental figures (and only with those that flux is not flowing through) - is this correct?

In figure 3 - the flux values have the potential to detract from what the authors are trying to illustrate - and in some situations they do not appear to be balanced (which is just a consequence of portraying a portion of the pathways and not the whole model). This is mirrored also in supplemental data where pathways being used look to be disconnected, but again, is due to the way that the visualization is presented. For readers new to metabolic modeling, this can be confusing.

In light of fig 3 - the discussion from lines 278-290 with flux values are not aligned with those presented in the figure. I understand that the supplemental figures are referred to here, but it seems that these should be complementary to each other.

Line 339: the authors state that fae1 is essential, yet, this is listed as a mutant that was generated.

Line 353: this does not hold true for liquid culture

It seems that figure 3 is one of the pivotal figures to the paper in pointing out that the CS model utilizes the H4F pathway preferentially and that this is no longer reversible as in the base model.

The resolution in figure 6 is not readable.

Reviewer #3 (Comments for the Author):

Firstly, I would like to apologise for the delay in providing this review.

This study presents a computational workflow for context-specific genome-scale model (CS-GSM) reconstruction, integrating COBRApy, EscherPy, and RIPTiDe tools via a BioUML-based GUI. The approach, tested on *Methylovibrio alcaliphilum* 20ZR, a methane-utilising microbe, effectively optimises metabolic models using -omics data and predicts flux distributions. The CS model aligns with experimental data, elucidating key metabolic pathways, such as the formaldehyde oxidation enzyme (Fae1-2), and revealing distinct metabolic features under La-dependent growth conditions. The findings demonstrate the workflow's potential for studying non-model microbes, significantly contributing to metabolic modelling in both fundamental and applied contexts.

In the revised manuscript, the authors have addressed all reviewer comments thoroughly. The manuscript is greatly improved compared to the initial submission.

My only minor suggestion is regarding Figure 1, which contains an overwhelming amount of information and could benefit from simplification to improve clarity.

Overall, I am satisfied with the revisions and recommend this manuscript for acceptance.

Response to reviewers

First of all, we would like to thank the reviewers for their critical comments and helpful suggestions.

Based on these comments and suggestions, we have made careful modifications to the original manuscript. The reviewer's comments are shown in black, followed by our responses in blue. The modifications made to the manuscript following the comments are marked in the revised manuscript and Supplementary materials.

Reviewer #1:

Q1: There are still inconsistencies in the naming conventions of *fae* - namely 1, 2, 3 or 1-2, 3.

R1: The names of genes are *fae1*, *fae1-2* and *fae3*. We have changed all names of *fae* to *fae1* in the maintext and Figure 5 as was requested.

Q2: In addition, there are inconsistencies between the legends and what is presented in figures. The line thickness only comes across in the supplemental figures (and only with those that flux is not flowing through) - is this correct?

R2: Thank you for the critical remark. We have corrected the legends in the maintext and in the Supplementary file.

Q3: In figure 3 - the flux values have the potential to detract from what the authors are trying to illustrate - and in some situations they do not appear to be balanced (which is just a consequence of portraying a portion of the pathways and not the whole model). This is mirrored also in supplemental data where pathways being used look to be disconnected, but again, is due to the way that the visualization is presented. For readers new to metabolic modeling, this can be confusing.

R3: Many thanks for your recommendation. We have updated Figure 3, removed the flux values, and highlighted the main difference between the original model and CS one to make it more clear for understanding. Complete data on fluxes distribution is provided via the gitlab project.

Q4: In light of fig 3 - the discussion from lines 278-290 with flux values are not aligned with those presented in the figure. I understand that the supplemental figures are referred to here, but it seems that these should be complementary to each other.

R4: Yes, Figure 3 is not aligned with the discussion from lines 278-290, because Figure 3 demonstrates the comparison of CS model including FAEii reaction with the original one, while the discussion in these lines and corresponding Supplementary Figures referred to this part of the maintext represent difference in distributions of fluxes between original model and CS one before addition of the FAEii reaction into the model.

Q5: Line 339: the authors state that *faei* is essential, yet, this is listed as a mutant that was generated.

R5: We would like to clarify the confusion. We only stated that the plasmids were constructed and introduced into an alternative host – *M. extroquens AM1Δfae1* for complementation studies.

“In this study three plasmids, harboring *fae1* (MEALZ_RS11875), *fae1-2* (MEALZ_RS04100) and *fae3* (MEALZ_RS07105) homologues from *M. alcaliphilum 20Z^R* were constructed and integrated into *M. extroquens AM1Δfae1*.”

We state on Page 14, L330-333 that “*We also constructed M. alcaliphilum 20Z mutants lacking fae1-2 or fae3; however, the mutants did not display any growth defects (data not shown). We were not able to generate fae1 knockouts, which indicates its essentiality for the growth.*”.

Q6: Line 353: this does not hold true for liquid culture

R6: That is correct. Only *fae1* required the AM1 phenotype in liquid culture. This is why we present the outcomes for both growth conditions. The significant discrepancies between the phenotypes might suggest some significant differences in the protein function. We elaborate on that in the Discussion part of the manuscript.

Q7: It seems that figure 3 is one of the pivotal figures to the paper in pointing out that the CS model utilizes the H4F pathway preferentially and that this is no longer reversible as in the base model.

R7: To emphasize the main differences between the original and CS models, we have updated Figure 3.

Q8: The resolution in figure 6 is not readable.

R8: We apologize for the unsuccessful transfer of the Figure 6 file. The phylogenetic analyses were re-done and the tree was redrawn. The modified file should represent the information in a readable format.

Reviewer #3

Q1: My only minor suggestion is regarding Figure 1, which contains an overwhelming amount of information and could benefit from simplification to improve clarity.

R1: Thank you for the suggestion. We have updated the Figure considering your recommendation.

Re: mSystems01105-24R1 (Modification and Analysis of Context-Specific Genome-Scale Metabolic Models: Methane-Utilizing Microbial Chassis as a Case Study)

Dear Dr. Marina G. Kalyuzhnaya:

The paper has addressed all the comments. I encourage the authors to make their codes available via Github as Dryad or a similar doi-generating depository. <https://journals.asm.org/open-data-policy>

Your manuscript has been accepted, and I am forwarding it to the ASM production staff for publication. Your paper will first be checked to make sure all elements meet the technical requirements. ASM staff will contact you if anything needs to be revised before copyediting and production can begin. Otherwise, you will be notified when your proofs are ready to be viewed.

Cover Image Submissions: If you would like to submit a potential Cover Image, please email a file and a short legend to mssystems@asmusa.org. Please note that we can only consider images that (i) the authors created or own and (ii) have not been previously published. By submitting, you agree that the image can be used under the same terms as the published article. Image File requirements: TIF/EPS, 7.5 inches wide by 8.25 inches tall (at least 2,250 pixels wide by 2,475 pixels tall), minimum 300 dpi resolution (600 dpi preferred), RGB, and no figure elements, e.g., arrows or panel labels. The legend should be a short description of the image, 1-2 sentences recommended. Please download and use this interactive template in Adobe to ensure that your proposed cover image meets our size requirements (<https://journals.asm.org/pb-assets/pdf-text-excel-files/ASM-Interactive-Sizing-Cover-Template-1715689791.pdf>).

We recognize that the video files can become quite large, so to avoid quality loss ASM suggests sending the video file via <https://www.wetransfer.com/>. When you have a final version of the video and the still ready to share, please send it to mSystems staff at mssystems@asmusa.org.

Sincerely,
Oleg Igoshin
Editor
mSystems